# Age-related changes in dermal collagen physical properties in human skin

**Tianyuan He, Gary J. Fisher, Ava J. Kim, Taihao Quan** [ID] *

Department of Dermatology, University of Michigan Medical School, Ann Arbor, Michigan, United States of America

* thquan@umich.edu

## Abstract

Collagen is the major structural protein in the skin. Fragmentation and disorganization of the collagen fibrils are the hallmarks of the aged human skin dermis. These age-related alterations of collagen fibrils impair skin structural integrity and make the tissue microenvironment more prone to skin disorders. As the biological function of collagen lies predominantly in its physical properties, we applied atomic force microscopy (AFM) and nanoindentation to evaluate the physical properties (surface roughness, stiffness, and hardness) of dermal collagen in young (25±5 years, N = 6) and aged (75±6 years, N = 6) healthy sun-protected hip skin. We observed that in the aged dermis, the surface of collagen fibrils was rougher, and fiber bundles were stiffer and harder, compared to young dermal collagen. Mechanistically, the age-related elevation of matrix metalloproteinase-1 (MMP-1) and advanced glycation end products (AGEs) are responsible for rougher and stiffer/harder dermal collagen, respectively. Analyzing the physical properties of dermal collagen as a function of age revealed that alterations of the physical properties of collagen fibrils changed with age (22–89 years, N = 18). We also observed that the reticular dermis is rougher and mechanically stiffer and harder compared to the papillary dermis in human skin. These data extend the current understanding of collagen beyond biological entities to include biophysical properties.

## Introduction

Skin is the largest organ of the body. Like all human organs, skin undergoes deleterious alterations because of the natural aging process. Additional damage is superimposed on this because of environmental exposure, such as ultraviolet light from the sun (Photoaging). At molecular levels, naturally aged and sun-induced photoaged human skin share common molecular features including accumulation of damaged dermal connective tissue collagen [1], which comprises the bulk of the skin. Fragmentation of collagen fibrils is the hallmark of skin dermal aging [1, 2]. Age-related alteration of the collagen is the major contributing factor to the clinical changes, such as fragile and wrinkled skin, the prominent clinical features of skin aging. Mechanistically, age-related alteration of collagen fibrils is largely driven by elevated matrix metalloproteinases (MMPs) [1, 3], which degrade collagen fibrils in the skin. These age-related progressive alterations of dermal collagen are intimately linked to the decline of skin functions, such as poor wound healing [4] and cancer development [5, 6].

**Data Availability Statement:** All relevant data are within the paper.

**Funding:** This work was supported by the National Institute of Health (R01ES014697 and

R01ES014697-03S1 to TQ). The funders had no role in study design, data collection and analysis, decision to publish, or preparation of the manuscript.

**Competing interests:** The authors have declared that no competing interests exist.

As we age, changes in the physical characteristics of dermal collagen become more noticeable. These alterations primarily result from degenerative processes, including the breakdown of collagen by proteases and non-enzymatic processes like collagen glycation, which lead to changes in its structure. Several studies have explored how aging affects the mechanical properties of different tissues, such as age-related increased stiffness of tendon, vascular, myocardial tissue, cartilage, and skin [7–9]. While age-related changes in the physical attributes of the human skin have been reported [7, 10–14], there is a lack of consensus primarily attributed to variations in measurement methodologies and experimental setups. Furthermore, a comprehensive understanding of the underlying mechanisms driving these age-related transformations in the skin dermal physical attributes remains elusive. Consequently, the connection between age and these property alterations remains unclear. It is worth noting that only a limited number of studies have delved into the impact of aging and age-related modifications on the dermal collagen at a nanoscale, such as the ruggedness of the dermal collagen fibrils. Collectively, the results from these studies lack consistency, making it difficult to definitively ascertain the link between aging and the physical properties of skin tissue. Thus, additional research is imperative to shed more light on this subject. Here we applied atomic force microscopy (AFM) and nanoindentation to investigate the physical properties of dermal collagen in young and aged human skin. We found that in aged human skin, the surface of collagen fibrils is rougher, and the collagen fiber bundles are stiffer and harder, compared to young dermal collagen. It is likely that age-related structural changes of the dermal collagen due to fragmentation and crosslinking of the collagen fibrils contribute to altered physical properties. These data provide insight into the physical properties of the damaged and disorganized collagen in aged human skin.

## Materials and methods

### Procurement of human skin samples *in vivo* and compliance with ethical standards

Young (25±5 years, N = 6) and aged (75±8 years, N = 6) volunteers were grouped according to age. Full-thickness human skin biopsies (4 mm) were obtained from the sun-protected buttock skin of each subject by punch biopsy and embedded in OCT. Volunteers were recruited and collected skin biopsies from August 2022 to January 2023. All skin samples were obtained under a protocol approved by the University of Michigan Institutional Review Board. All volunteers provided written informed consent.

### Atomic force microscopy (AFM) imaging

Cryo-sections (15 μm) were prepared from OCT-embedded skin and attached to the microscope cover glass (1.2 mm diameter, Fisher Scientific Co., Pittsburgh, PA). These AFM samples were allowed to air dry for at least 24 hours before AFM analysis. Nanoscale collagen AFM images were obtained in air by Dimension Icon AFM system (Bruker-AXS, Santa Barbara, CA, USA) using a silicon AFM probe (PPP-BSI, force constant 0.01–0.5N/m, resonant frequency 12-45kHz, 10-nm-radius, NANOSENSORS™, Switzerland). AFM images of the collagen fibrils were acquired using ScanAsyst mode, which is an optimized PeakForce Tapping technique that enables users to create the high resolution AFM images. ScanAsyst mode can automatically and continuously monitor image quality and make appropriate parameter adjustments. AFM images were obtained from the reticular and papillary dermis with a 512 × 512-pixel resolution (per skin section/subject, 5×5μm scan size)The surface roughness of the images was quantified using Nanoscope Analysis software (Nanoscope Analysis

v120R1sr3, Bruker-AXS, Santa Barbara, CA, USA). The surface roughness of the images was quantified from raw images without modification, such as clean images, flattening, filtering, and plane fitting. Quantification of the surface roughness was obtained from a total of 72 AFM images from each group (6 images/subject × total 12 subjects = 72 images/group). AFM was conducted at the Electron Microbeam Analysis Laboratory (EMAL), University of Michigan College of Engineering.

## Nanoindentation measurements

OCT-embedded human skin biopsies were sectioned (100μm) and were attached to the microscope cover glass (1.2 mm diameter, Fisher Scientific Co., Pittsburgh, PA). Mechanical properties (stiffness/hardness and Young's modulus) were measured by nanoindentation by using a NanoIndenter II (Agilent Technologies, Santa Clara, CA) in the constant displacement rate loading mode with a three-sided pyramidal diamond tip. A fused quartz sample with known hardness and Young's modulus values was used as a reference sample. The maximum indentation displacement was maintained at 2000 nm. The method used to calculate the hardness and the stiffness modulus was based on established methods [15, 16]. A total of six indents per skin section/subject were obtained from the papillary and reticular dermis (Fig 2A). Quantification of the skin dermal mechanical properties was obtained from a total of 36 indents from each group (6 indents/subject × total 6 subjects = 36 indents/group).

## RNA isolation and quantitative real-time RT-PCR

To determine gene expression in the dermis, the dermis was prepared by cutting off the epidermis at a depth of 1 mm by cryostat (Fig 3A). Dermal tissue RNA was extracted using a RNeasy micro kit (Qiagen, Gaithersburg, MD, USA) according to the manufacturer's protocol. cDNA template for PCR amplification was prepared by reverse transcription of total RNA (200 ng) using a TaqMan Reverse Transcription kit (Applied Biosystems, Carlsbad, CA, USA). Real-time PCR quantification was performed on a 7300 Sequence Detector (Applied Biosystems, Carlsbad, CA, USA) using TaqMan Universal PCR Master Mix Reagents (Applied Biosystems, Carlsbad, CA, USA). PCR primers were purchased from RealTimePrimers.com (Real Time Primers, LLC, Elkins Park, PA, USA). Target gene mRNA expression levels were normalized to the housekeeping gene 36B4 as an internal control for quantification.

## Human skin organotypic culture

Human skin in organ culture has been extensively used in the past and the protocol used here was virtually identical to that described previously in our department [17]. Briefly, human skin punch biopsies (2mm) were obtained from the hip skin of volunteers (21–30 yrs). Skin samples were incubated in a 24-well dish (one tissue piece per 500 μl of culture medium). The culture medium consisted of Keratinocyte Basal Medium (KBM) (Lonza, Walkersville, MD), supplemented with $CaCl_2$ to a final concentration of 1.4 mM. For MMP-1 digestion, skin samples were treated with activated human MMP-1 (150 ng/ml, Calbiochem, CA) [18] for 24 hours. Prior to use, MMP-1 was activated by trypsin treatment (1.25ng per 100ng MMP-1) for one hour at 37˚C. Trypsin activity was inhibited by the addition of trypsin inhibitor (12.5ng). MMP-1 was inactivated by adding EDTA (10mM) followed by the addition of $CaCl_2$ (15mM), then the gel was washed three times with fresh DMEM. To confirm MMP-1 activity, intact collagen was exposed to MMP-1 *in vitro*, and intact and fragmented collagens were resolved in 10% SDS-polyacrylamide gel (Invitrogen, CA) and visualized by staining with SimplyBlue. To validate the collagen degradation facilitated by MMP-1, the skin samples were subjected to staining using collagen hybridizing peptides (3HGelix, Salt Lake City, UT. USA) according to

the manufacturer's instructions. These peptides are designed to selectively detect degraded collagen [19]. For *in situ* glycation, skin samples were exposed to ribose (300 mM, Sigma-Aldrich, Saint Louis, MO, USA) for seven days. To confirm collagen glycation, skin samples underwent pepsin digestion (0.2 mg/mL of pepsin in 0.5 M acetic acid, Sigma-Aldrich, Saint Louis, MO, USA) at a temperature of 37°C overnight. Following the digestion process, the pH was adjusted to 7.0 using 0.5 M sodium hydroxide and then centrifuged at 10,000 g for 5 minutes. The resulting supernatant, which contained the digested collagen, was subjected to fluorescence measurement to determine the total advanced glycation end-products (AGEs) present. This measurement was carried out with an excitation wavelength of 370 nm and an emission wavelength of 440 nm ($\lambda$ex 370 nm/$\lambda$em 440 nm), as previously described [20, 21]. At the end of the incubation period, the tissues were embedded in OCT and processed for AFM and nanoindentation analysis.

## Histomorphometry analysis

Sections (5 μm) from formalin-fixed skin samples were stained with hematoxylin/eosin and Sirius red. To analyze the morphological characteristics of dermal collagen fibers, the sections were assessed using three criteria indicating changes in dermal fibers: (i) spacing between fibers, (ii) thinness of fibers, and (iii) arrangement disorder of fibers. Each of these parameters was evaluated using a numerical scale ranging from 1 to 9 [22]. A score of 1 indicated a favorable condition, while a score of 9 indicated the least favorable state. This numeric framework enables a quantitative assessment, facilitating a comprehensive comparison of traits concerning dermal collagen fibers in young and aged human skin.

## Charts and statistics

The data were organized in Microsoft Excel 365, and then transferred into GraphPad Prism (v.8) for statistical analysis and graph generation. All data are represented as Mean± SEM. Statistical analysis was performed using GraphPad Prism (v.8) with unpaired two-sided Student's -*t*-tests, one-way analysis of variance (ANOVA) with Tukey's method for multiple comparisons, or Kruskal-Wallis test with Dunn's multiple comparisons test. Statistical significance was defined as $P < 0.05$. All experiments were repeated a minimum of three times unless otherwise stated.

## Results

### Rougher collagen fibril surface in aged dermis compared to young dermis

We first assessed histological images of the skin biopsies (six young 25±5 years and six aged 75 ±6 years) used in the current study by H&E and Sirius red staining (Fig 1A). We observed that in aged human dermis, there are increased dermal collagen connective tissue abnormalities, as indicated by increased thinning of collagen fiber bundles, increased space between collagen fiber bundles, and increased disorganization of fiber bundles (Fig 1B). We analyzed nanostructures of the dermal collagen by AFM (Fig 1C). In the young dermis, intact collagen fibrils are abundant, tightly packed, well-organized, and display characteristic d-bands. (Fig 1D left panel). In contrast, collagen fibrils are fragmented and disorganized in the aged dermis (Fig 1D right panel). To measure the integrity and organization of collagen fibrils, we quantified the roughness of the collagen fibril surface based on the height profiles of the collagen fibril cross-section. Surface roughness is a component of surface texture measured by Ra, which is calculated as the mean deviation of height over the entire measured area. Large deviations indicate a rough surface (indication of collagen fragmentation and disorganization), while

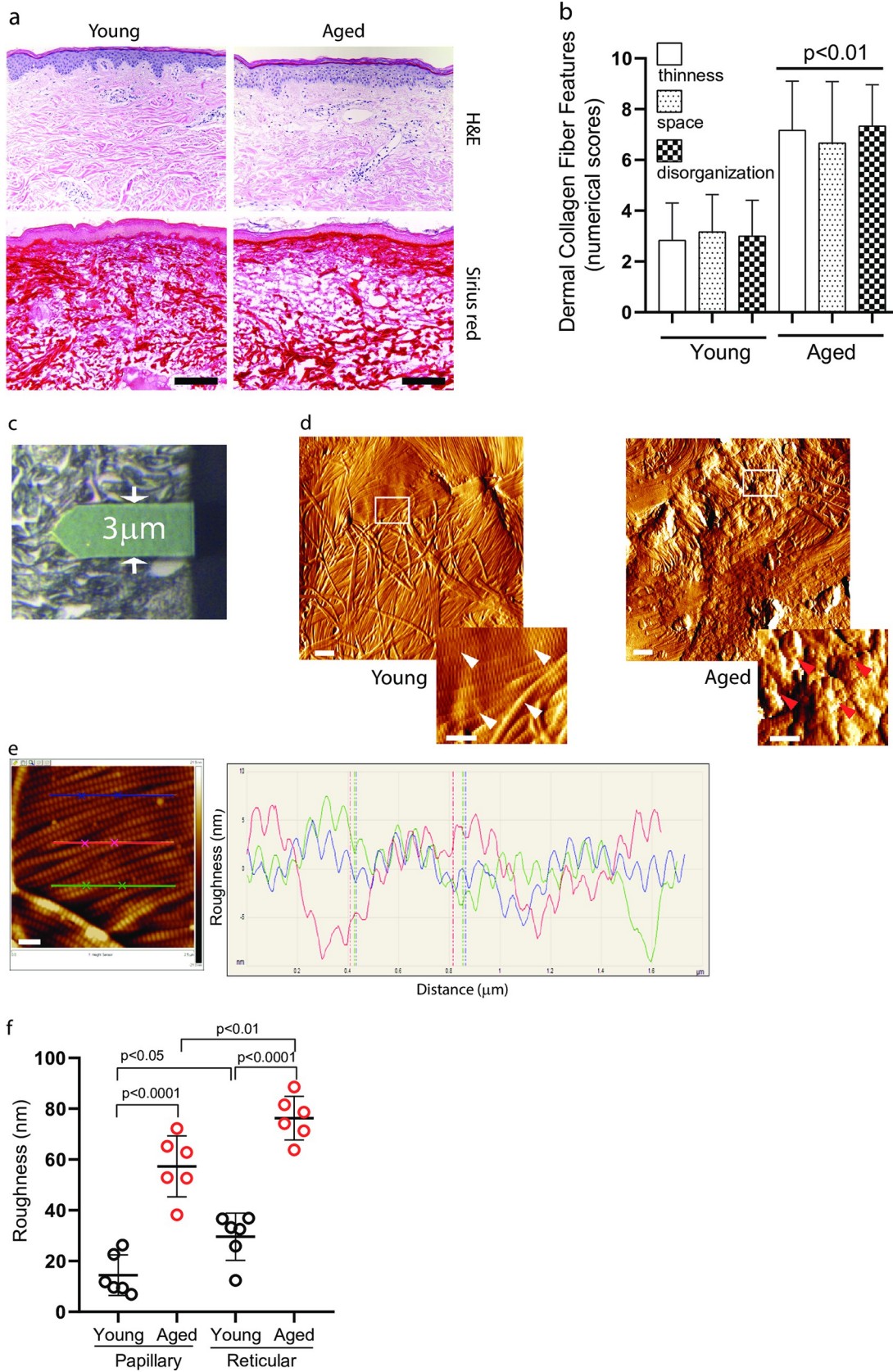

**Fig 1. Rougher collagen fibril surface in aged dermis.** (a) Histological image of young (left, 28 years) and aged (right, 82 years) human skin. Skin sections are obtained from sun-protected hip and stained with H&E (upper) and Sirius red (bottom). Images are representative of six young (25±5 years) and six aged (75±8 years) subjects. Bar = 100μm. (b) Skin dermal collagen alterations in aged human skin. The morphological characteristics of dermal collagen fibers were assessed based on three criteria reflecting alterations in dermal collagen fibers: (i) inter-fiber spacing, (ii) fiber thinness, and (iii) disorder in fiber arrangement. Each of these factors was assessed using a numerical scale that ranged from 1 (minimally evident) to 9 (highly evident).Results are expressed as the mean ± SEM. (c) Representative bright field image shows the AFM cantilever positioned on the dermis. AFM images were obtained from the reticular and papillary dermis (rectangles, 5×5μm scan size). (d) AFM nanoscale images of the collagen fibrils from young (left, 28 years) and aged (right, 82 years) human skin. White and red arrows indicate intact and fragmented collagen fibrils, respectively. Images are representative of six young and aged subjects. Bar = 1μm (insert = 500nm). (e) Representative image for quantification of collagen fibril surface roughness. Lateral dimension is 2.5x2.5 μm$^2$. Height is given in black and white brightness. The lines indicate cross sections that are displayed by graph. Each line (blue, red, and green) height fluctuations in the graph indicate corresponding collagen surface roughness of the image. Bar = 200nm. (f) The roughness of collagen fibril surface is increased in aged human skin. The roughness of collagen fibrils was analyzed using Nanoscope Analysis software (Nanoscope_Analysis_v120R1sr3, Bruker-AXS, Santa Barbara, CA). Each group comprised six subjects. Results are expressed as the mean ± SEM.

small deviations denote a smooth surface (intact and well-organized collagen fibrils). Fig 1E shows a typical topographical image (left) and corresponding height profile (right) of the dermal collagen fibril surface cross-section. Considering that the roughness of the papillary and reticular dermis could be different, we measured the roughness in both the papillary and reticular dermis. Quantitative analysis indicated that the surface of dermal collagen fibrils in the aged dermis has a much higher roughness in both papillary and reticular dermis, compared to the young dermis (Fig 1F). These findings demonstrate that the surface of collagen fibrils in aged dermis is rougher/disorganized, compared to young dermal collagen fibrils. We also observed that the surface of collagen fibrils in the reticular dermis is rougher compared to the papillary dermis in both young and aged human skin.

## Stiffer and harder collagen fiber bundles in aged dermis compared to young dermis

Next, we measured the mechanical properties of the collagen fiber bundles using a nanoindenter by measuring the two key biomechanical properties, stiffness and hardness. Considering that the physical properties of the papillary and reticular dermis could be different, a total of 6 indents (3 from papillary, 3 from reticular dermis) throughout the dermis were measured (Fig 2A upper panel). Fig 2A (bottom panel) shows a typical load–displacement curve obtained in the present study. Interestingly, collagen fiber bundles in aged papillary and reticular dermis revealed a higher stiffness (Fig 2B) and hardness in both papillary and reticular dermis (Fig 2C), compared to young dermis. In the aged reticular dermis, both stiffness (0.07mN/nm ±0.005) and hardness (6.1GPa±0.5) of the collagen fiber bundles were increased by 175% and 145%, respectively, compared to young skin dermis (stiffness: 0.04mN/nm±0.002, hardness: 4.2GPa±0.4). Taken together, these findings demonstrate that collagen fiber bundles in the aged dermis are stiffer and harder, compared to young dermis. We also observed that the reticular dermis is stiffer and harder compared to the papillary dermis in both young and aged human skin.

## Elevated matrix metalloproteinase-1 contributes to rougher collagen fibril surface in aged dermis

Matrix metalloproteinases (MMPs) are responsible for collagen fibril fragmentation. We previously reported that MMP-1 is elevated in aged human skin [3]. We tested whether elevated MMP-1 results in altered physical properties of the collagen fibrils in the aged dermis. To this end, we first evaluated MMP-1 mRNA expression from the skin samples used in this study

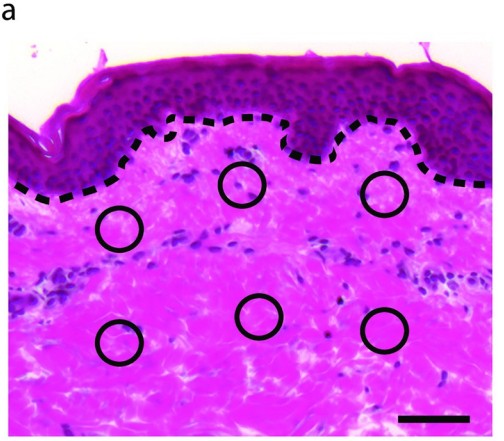

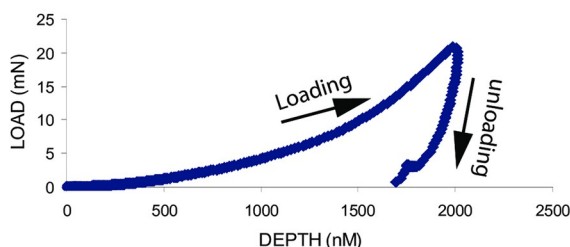

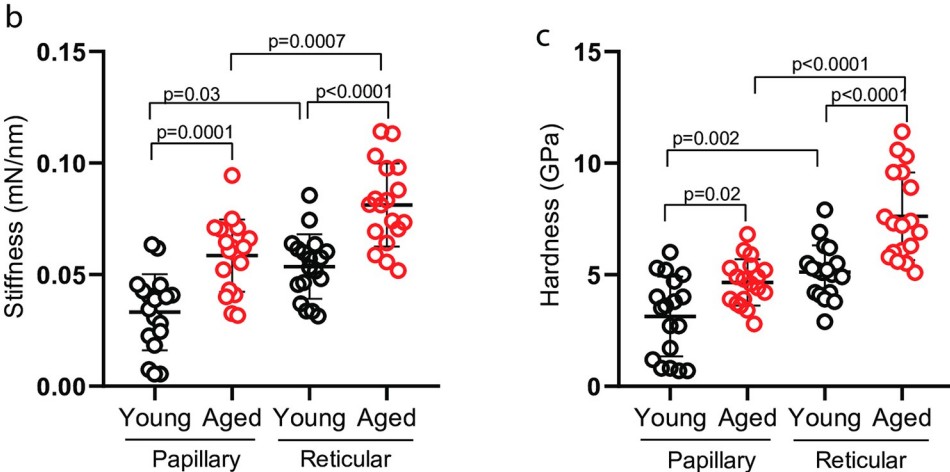

**Fig 2. Stiffer and harder collagen fiber bundles in aged dermis.** (a) Representative image for quantification of mechanical properties by Nanoindentation. Total of six indents (3 from papillary, 3 from reticular dermis), as indicated by circles, per skin section were obtained. The graph represents typical load (mN) and displacement (penetration depth, nm) curve (see Methods for details). Dot lines indicate epidermal and dermal junction. Bar = 100μm. (b) Increased collagen fiber bundle stiffness in aged human skin. (c) Increased collagen fiber bundle hardness in aged human skin. Stiffness and hardness were quantified using a NanoIndenter II (Agilent Technologies, Santa Clara, CA), as described in Methods. Each group comprised six subjects. Results are expressed as the mean ± SEM.

(young, 25±5 years, N = 6; aged 75±6 years, N = 6). To determine MMP-1 expression in the dermis, the skin dermis was prepared by cutting off the epidermis at a depth of 1 mm by cryostat (Fig 3A). We confirmed that significantly elevated MMP-1 mRNA expression in the aged dermis, compared to the young dermis (Fig 3A). Treatment of rat tail collagen with human MMP-1 (hMMP-1) generated one-quarter and three-quarter length collagen fragments (Fig 3B and S1 Raw image), which are characteristic of MMP-1 activity [6]. Confirmation of dermal collagen degradation was achieved by staining skin specimens with collagen hybridizing peptides (3HGelix, Salt Lake City, UT, USA) (Fig 3C), which exhibit a specific affinity for fragmented collagen [23]. Treatment of young human dermis with hMMP-1 by organ culture resulted in significant alterations of collagen fibrils (Fig 3D), as observed in aged human dermis (Fig 1D). Interestingly, we found that treatment of young human skin dermis with hMMP-1 resulted in a rough surface of the collagen fibrils (Fig 3E), while no significant change was observed in stiffness (Fig 3F) and hardness (Fig 3G). These data suggest that elevated MMP-1 contributes to rougher collagen fibril surface in the aged dermis.

## AGE-mediated collagen-crosslinking contributes to stiffer and harder collagen fiber bundles

Next, we investigated the impact of AGEs on altered physical properties of collagen fibrils in the aged dermis. AGEs are known to accumulate in human aging skin, cross-link collagen, and thus are capable of changing the tissue's mechanical properties [24, 25]. We explored the impact of ribose-mediated collagen crosslinking on the mechanical properties of dermal collagen fibrils. Ribose glycation was achieved by incubation of young human skin punch biopsies with ribose in organ culture. The dermal collagen glycation was confirmed by fluorescence measurement ($\lambda$ex 370nm/$\lambda$em 440nm) [20, 21] (Fig 4A). AFM imaging showed no significant morphological change in collagen fibrils after ribose-mediated crosslinking (Fig 4B). However, cross-linking of young skin with ribose resulted in a significant increase in stiffens (Fig 4C) and hardness (Fig 4D) of the collagen fiber bundles, as observed in the aged dermis (Fig 2), while no significant change was observed in the roughness of collagen fibril surface (Fig 4E). These data suggest that AGEs contribute to stiffer and harder collagen bundles in the aged dermis.

## Age-related changes in collagen physical properties in human skin

The above data demonstrate that the physical properties of collagen fibrils are significantly altered in the aged dermis. Next, we explored the changes in the physical properties of dermal collagen as a function of age (age range 25–89 years, total 18 subjects). We observed that the physical properties of collagen; roughness (Fig 5A), stiffness (Fig 5B), and hardness (Fig 5C), are positively correlated with age, suggesting age-related changes in the physical properties of the collagen fibrils in human skin.

## Discussion

Collagen is the most abundant mammalian protein in the body and contributes significantly to the structural and mechanical properties of tissue. It is also well-known that the biological functions of cells are dependent on the surrounding tissue's mechanical properties [8]. There is a fundamental need to understand the physical properties of human skin, the largest organ of the body. In this study, we have applied AFM imaging and nanoindentation techniques to evaluate the physical properties of collagen fibrils in the human dermis. Our data reveal that fragmented and disorganized collagen in aged human skin is rougher, mechanically stiffer, and harder, compared to intact and well-organized collagen in young dermis. Furthermore,

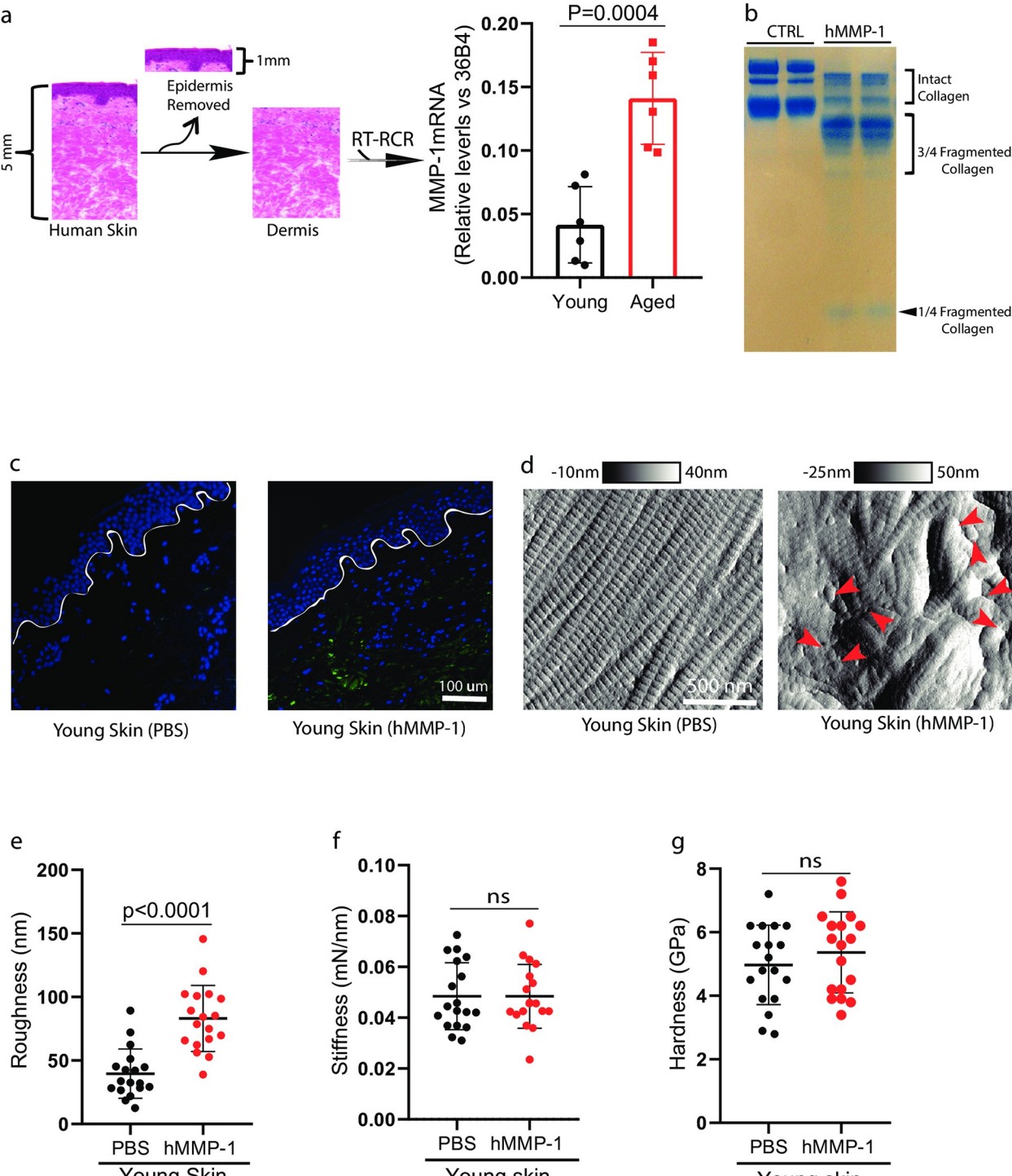

**Fig 3. Age-related elevation of MMP-1 causes rougher surface of the collagen fibrils.** (a) Elevated MMP-1 mRNA expression in aged human dermis. Schematic representation of the dissection of human skin dermis by cutting off epidermis at a depth of 1 mm by cryostat (left). Total RNA was extracted from the dermis. MMP-1 and 36B4 (internal normalization control) mRNA levels were quantified by real-time RT-PCR. Each group comprised six subjects. Mean± SEM. (b) Treatment of rat tail collagen with human MMP-1 generated one quarter and three-quarter length collagen fragments. Intact and fragmented collagens were resolved in 10% SDS-polyacrylamide gel and visualized by staining with SimplyBlue. (c and d) Young human skin was treated with human MMP-1 in organ culture resembles aged dermis. Collagen degradation was confirmed by staining using collagen hybridizing peptides (c) and visualized through AFM imaging (d). Lines indicate epidermal dermal junction. Arrow heads indicate fragmented collagen fibrils. Images are representative of six subjects. Young dermis collagen (d) roughness, (e) stiffness, and (f) hardness after hMMP-1 treatment. N = 6, results are expressed as the mean ± SEM.

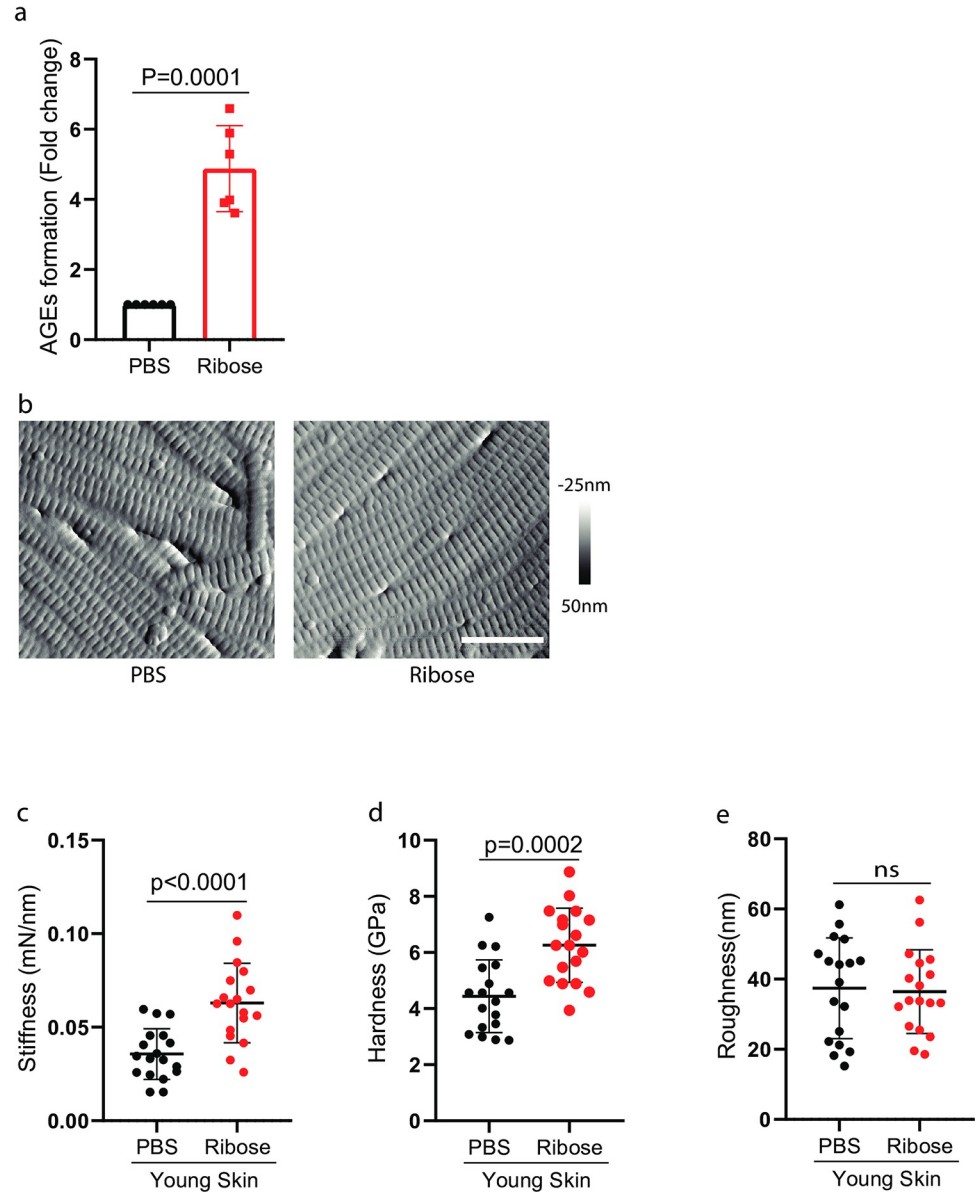

**Fig 4. AGE-medicated collagen-crosslinking contributes to stiffer and harder collagen fiber bundles.** (a) Young human skin was treated with ribose in organ culture. The glycation of dermal collagen was verified using fluorescence measurements (λex 370nm/λem 440nm) (a) and visualized through AFM imaging (b). Images are representative of six subjects. Bar = 500nm. Young dermis collagen (c) stiffness, (d) hardness, and (e) roughness after ribose-mediated collagen-crosslinking. N = 6, results are expressed as the mean ± SEM.

dermal physical properties of collagen fibrils change with age, suggesting that aging affects the physical properties of the skin dermis.

Elevation of MMP-1 and consequent dermal collagen fibril fragmentation in aged human skin are well characterized [3]. Evidence suggests that MMP-1-mediated cumulative collagen damage is a major contributor to the phenotype of aged human skin [1]. Although the biological function of elevated MMP-1 in dermal aging is widely recognized, the relationship between MMP-1 and collagen biophysical properties has received little attention. Our findings demonstrate that MMP-1-mediated collagen fragmentation drives rougher collagen fibril surfaces in

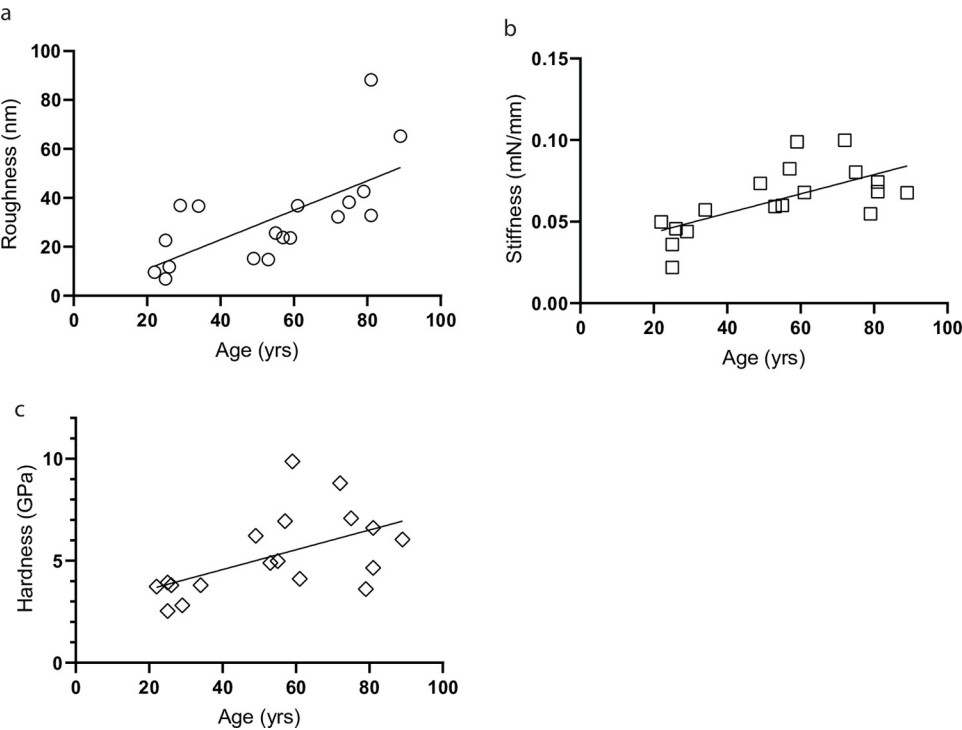

**Fig 5. Age-related alteration of the dermal collagen physical properties in human skin.** The roughness collagen fibril surface and mechanical properties (stiffness and hardness) were measured by AFM and nanoindentation, respectively (see Methods for details). Each data point is representative of each subject (age range 25–89 years, total 18 subjects). Collagen (a) roughness, (b) stiffness, and (c) hardness are plotted as a function of age.

aged human skin. It is not surprising that in general, fragmented, and disorganized collagen fibrils could create a rougher surface. Fragmentation of the collagen fibrils could result in disintegration and disorder of the collagen fiber bundles, which may influence collagen stiffness and hardness. However, we find that fragmented/disorganized-rougher collagen fibrils are not stiffer and harder, compared to intact well-organized collagen.

Mechanistically, AGE-mediated collagen-crosslinking contributes to the stiffer and harder dermis. It is well-known that AGEs accumulate with age [10, 26]. The accumulation of AGEs occurs in tissues with a low turnover rate such as skin dermal collagen. Our data suggest that AGE-mediated collagen-crosslinking can lead to stiffer and harder skin dermis. The essential functions of the human skin depend on the mechanical properties of the dermis that provide skin with strength, flexibility, and elasticity. Skin homeostasis is intrinsically linked to the mechanical properties of the skin dermis. Age-related changes in dermal mechanical properties may have important implications on how the skin behaves in both physiological and pathological circumstances. Changes in dermal mechanical properties may have significant implications on age-related skin disorders such as impaired wound healing, which is largely controlled by dermal collagen mechanical properties. Additional work is warranted to understand how changes in the mechanical properties of the dermis influence age-related skin disorders. We also observed that the reticular dermis is rougher and mechanically stiffer and harder compared to the papillary dermis in human skin. It is known that compared to the papillary dermis, the reticular dermis is more acellular and has thicker and denser collagen, which may largely contribute to the differences in the physical properties between the reticular and papillary dermis.

The aging process prominently displays alterations in the physical attributes of skin tissue [7, 13]. Nonetheless, a multitude of research studies have presented varied and conflicting outcomes. For instance, when comparing our own research conducted on buttock skin with prior studies, Ahmed et al documented a decrease in the Young's modulus of collagen fibrils in forearm skin as a result of aging [12]. Several factors might contribute to these divergent results, including dissimilarities in skin anatomy across various sites, yielding distinct outcomes in terms of mechanical properties [7]. Prior investigations have revealed that the elastic modulus of skin tissue is higher on the volar forearm compared to the dorsal forearm and palm [27]. Conversely, another study suggested that Young's modulus of chest skin tends to be lower than that of skin on the finger, forearm, and hand [28]. Even within the same skin section, we found that the collagen fibrils within the reticular dermis exhibit a more uneven surface texture and greater mechanical stiffness and hardness in comparison to the collagen fibrils found in the papillary dermis of human skin. The collagen fibrils in the reticular dermis are notably thicker and exhibit a more intricate arrangement compared to those in the papillary dermis [29]. It has become evident that the physical characteristics of these dermal collagen fibrils are influenced by factors such as their diameter and orientation [13]. We also recognize that the current study has several drawbacks, such as the mechanical properties of air-dried skin dermis should be different from that of naturally-accruing wet skin dermis. It is worth mentioning that we failed in our attempt to measure mechanical properties using fresh/wet human skin samples due to technical challenges. It is also unknown how the alignment and density of the dermal collagen fibrils influence the accurate measurement of the true mechanical properties of the skin dermis. Clearly, the contribution of the collagen alignment and density to the dermal mechanical properties will need to be taken into account to clarify this question.

Collagen-rich dermal ECM serves many purposes, including mechanical strength and resiliency, physical support for appendages, blood vessels, nerves, and lymphatics, a dynamic scaffold for the attachment of cells, and a repository and regulator of potent biological mediators (growth factors, cytokines, chemokines, matricellular proteins, etc.). As such, the dermal microenvironment provides not only chemical signals but also inputs of a mechanical nature. The mechanical property of the tissue microenvironment is critically important in controlling the cell's fundamental functions [30]. For example, matrix stiffness can control the differentiation of mesenchymal stem cells into distinct lineages and tumorigenesis. As such, age-related alteration of dermal collagen physical properties may have a significant impact on dermal cellular functions.

In general, cellular biomechanics are increased and cells are more active under conditions of stiffer and harder tissue microenvironment [31]. This is true in the case of dermal fibroblasts [32]. We have reported that dermal fibroblasts' mechanical properties are enhanced and the ECM synthetic activity is much more active in stretched versus relaxed collagen lattices, as well as growing in plastic dishes versus collagen gels [33, 34]. We reported that an important characteristic feature of aged dermal fibroblasts is loss of mechanical tension and cell shape, which are critically important in fibroblast function [32, 35]. One interesting implication of the current manuscript is that in human skin *in vivo*, aged dermal fibroblasts may be unable to sense stiffer and harder mechanical cues from the collagen microenvironment due to the loss of cell-ECM communications caused by fragmented collagen fibrils. Therefore, it is conceivable that although aged dermal fibroblasts reside in stiffer and harder collagen environments, their cellular biomechanics are reduced due to age-related fragmented collagen fibrils being unable to provide the changes of mechanical cue to the cell.

In summary, we report that naturally aged human skin dermis is rougher and mechanically stiffer and harder, largely due to biological changes in dermal collagen such as increased MMP-1-mediated collagen fragmentation and AGEs-mediated crosslinking. These data

provide useful information for our understanding of the nanostructural and biophysical properties of the damaged and disorganized collagen fibrils in aged human skin.

## Supporting information

**S1 Raw image. Original, uncropped electrophoresis gel picture underlying Fig 3b from the main text.**
(PDF)

## Acknowledgments

We thank Suzan Rehbine for the procurement of tissue specimens, and Diane Fiolek for administrative assistance.

## Author Contributions

**Conceptualization:** Taihao Quan.

**Data curation:** Tianyuan He, Gary J. Fisher, Ava J. Kim, Taihao Quan.

**Formal analysis:** Tianyuan He, Gary J. Fisher, Taihao Quan.

**Funding acquisition:** Taihao Quan.

**Investigation:** Tianyuan He, Gary J. Fisher, Ava J. Kim, Taihao Quan.

**Methodology:** Gary J. Fisher, Taihao Quan.

**Project administration:** Taihao Quan.

**Resources:** Taihao Quan.

**Software:** Taihao Quan.

**Supervision:** Taihao Quan.

**Validation:** Gary J. Fisher, Taihao Quan.

**Visualization:** Taihao Quan.

**Writing – original draft:** Taihao Quan.

**Writing – review & editing:** Gary J. Fisher, Ava J. Kim, Taihao Quan.

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
