## [Decision Letter · Decision Letter 0]

15 Aug 2023

PONE-D-23-19389Age-related changes in dermal collagen physical properties in human skinPLOS ONE

Dear Dr. Quan,

This is to inform you that your manuscript "Age-related changes in dermal collagen physical properties in human skin" requires a minor revision. Your manuscript was reviewed by expert referees who have made a number of recommendations regarding the suitability of your paper for publication in the PLOS ONE. The reviewers’ comments are provided below for your assistance in revising the paper. Therefore, we invite you to submit a revised version of the manuscript that addresses the points raised during the review process.

We look forward to receiving your revised manuscript.

Kind regards,

Nileshkumar Dubey

Academic Editor

PLOS ONE

“This work was supported by the National Institute of Health to TQ.”

“This work was supported by the National Institute of Health (R01ES014697 and R01ES014697- 03S1 to TQ), and the Department of Dermatology, University of Michigan.”

“This work was supported by the National Institute of Health to TQ.”

7. We note that Figures 2, 3 and 4 in your submission contain copyrighted images. All PLOS content is published under the Creative Commons Attribution License (CC BY 4.0), which means that the manuscript, images, and Supporting Information files will be freely available online, and any third party is permitted to access, download, copy, distribute, and use these materials in any way, even commercially, with proper attribution. For more information, see our copyright guidelines: http://journals.plos.org/plosone/s/licenses-and-copyright.

1. You may seek permission from the original copyright holder of Figures 2, 3 and 4 to publish the content specifically under the CC BY 4.0 license.

Reviewers' comments:

Reviewer's Responses to Questions

**Comments to the Author**

1. Is the manuscript technically sound, and do the data support the conclusions?

Reviewer #1: Yes

Reviewer #2: Yes

2. Has the statistical analysis been performed appropriately and rigorously? 

Reviewer #1: Yes

Reviewer #2: Yes

3. Have the authors made all data underlying the findings in their manuscript fully available?

Reviewer #1: Yes

Reviewer #2: Yes

4. Is the manuscript presented in an intelligible fashion and written in standard English?

Reviewer #1: Yes

Reviewer #2: Yes

5. Review Comments to the Author

Reviewer #1: Authors have drafted interesting piece of work which well aligned and well addressed.

Data of manuscript has been well articulated and hypothesis has well explained which fluently understandable by readers.

1. I recommend authors to go through entire manuscript once more for minor grammatical errors. I recommend to1

change term Organ culture to Organotypic culture, to show the skin biopies were cultured and maintained under in vitro conditions.

2. Authors are recommended toreplace old citation and cite recently published articles and to stick max 5 years old references only. ( 2018-2023).

Reviewer #2: Authors tried to clarify the age-related changes in dermal collagen physical properties (surface roughness, stiffness, and hardness) using atomic force microscopy (AFM) and nanoindentation. They found that in the aged dermis, the surface of collagen fibrils was rougher, and fiber bundles were stiffer and harder, compared to young dermal collagen and that the age-related elevation of matrix metalloproteinase-1 (MMP-1) and advanced glycation end products (AGEs) might be responsible for rougher and stiffer/harder dermal collagen, respectively. These findings are interesting but there are some points to improve this paper.

Comment 1

Although some studies have already been carried out to clarify the age-related changes in dermal collagen physical properties by using atomic force microscopy (AFM) and nanoindentation, authors did not describe these works in the introduction, Authors should describe them and emphasize the importance of this work.

Comment 2

In histology and morphometry (Page7), a scale of 1-9 for each parameter was used for the analysis. However, they did not describe the details of the difference of 1 to 9 scales. To understand the importance of Fig. 1b, they should show the differences of scales in thinness, space, and disorganization of dermal collagen fibers.

Comment 3

In Figs 3 and 4, in addition to AFM data, skin section data should be necessary to be shown as the effects of MMP-1 digestion and ribose crossing on the dermis. Moreover, for Fig. 4, the presence of a glycated collagen matrix should be shown by histological immunostaining.

Comment 4

In the paper, published in International Journal of Nanomedicine 2017:12 3303–3314, “Combining nano-physical and computational investigations to understand the nature of “aging” in dermal collagen”, an age-related decrease in the Young’s modulus of the transverse fibril (from 8.11 to 4.19 GPa in young to old volunteers, respectively, P,0.001) was reported. Authors should discuss the discrepancy.

Comment 5

In Figs 3c and 4c, the spelling of punch in “Young skin puch biopsy” should be corrected.

6. PLOS authors have the option to publish the peer review history of their article (what does this mean?). If published, this will include your full peer review and any attached files.

Reviewer #1: **Yes: **HARISH KIRAN HANDRAL

Reviewer #2: No

---

## [Author Response · Author response to Decision Letter 0]

14 Sep 2023

PONE-D-23-19389

TITLE: Age-related changes in dermal collagen physical properties in human skin

We appreciate the time and effort of the reviewers to provide their thoughtful comments regarding our manuscript. 

In response to their suggestions, we conducted additional experiments (depicted in Fig 3cs and Fig 4a) and revised the manuscript. 

Changes made to the manuscript are marked in the text. 

Point-by-point responses to reviewers’ comments are below.

Reviewer #1: 

Authors have drafted interesting piece of work which well aligned and well addressed.

Data of manuscript has been well articulated and hypothesis has well explained which fluently understandable by readers.

1. I recommend authors to go through entire manuscript once more for minor grammatical errors. I recommend to 1 change term Organ culture to Organotypic culture, to show the skin biopies were cultured and maintained under in vitro conditions.

Response: “Organ” culture changed to “Organotypic culture”, also corrected any grammatical errors. 

2. Authors are recommended to replace old citation and cite recently published articles and to stick max 5 years old references only. (2018-2023).

Response: The citations have been updated (please refer to the "References" section), yet certain references remained unaltered due to the lack of more up-to-date sources.

Reviewer #2: 

Authors tried to clarify the age-related changes in dermal collagen physical properties (surface roughness, stiffness, and hardness) using atomic force microscopy (AFM) and nanoindentation. They found that in the aged dermis, the surface of collagen fibrils was rougher, and fiber bundles were stiffer and harder, compared to young dermal collagen and that the age-related elevation of matrix metalloproteinase-1 (MMP-1) and advanced glycation end products (AGEs) might be responsible for rougher and stiffer/harder dermal collagen, respectively. These findings are interesting but there are some points to improve this paper.

Comment 1

Although some studies have already been carried out to clarify the age-related changes in dermal collagen physical properties by using atomic force microscopy (AFM) and nanoindentation, authors did not describe these works in the introduction, Authors should describe them and emphasize the importance of this work.

Response: By revisiting the Introduction section, we incorporated and deliberated upon the content of these papers (refer to the "Introduction" and “References” sections for more information).

Comment 2

In histology and morphometry (Page7), a scale of 1-9 for each parameter was used for the analysis. However, they did not describe the details of the difference of 1 to 9 scales. To understand the importance of Fig. 1b, they should show the differences of scales in thinness, space, and disorganization of dermal collagen fibers.

Response: We apologize for any confusion related to the scales utilized in our histological analysis. We have made revisions to the histological analysis scales and have provided a corresponding reference. Detailed information regarding these scales is available in the "Materials and Methods" section within the "Histomorphometry Analysis" category. Additionally, you can find supplementary information in the figure 1 legend.

Comment 3

In Figs 3 and 4, in addition to AFM data, skin section data should be necessary to be shown as the effects of MMP-1 digestion and ribose crossing on the dermis. Moreover, for Fig. 4, the presence of a glycated collagen matrix should be shown by histological immunostaining.

Response: Following the suggestions, we carried out additional experiments to verify the processes of MMP-1 induced collagen degradation (as shown in Fig 3c) and ribose-induced collagen glycation (depicted in Fig 4a). We would like to inform you that our endeavor to perform AEGs immunostaining was unsuccessful due to the issues with tissue freshness (new fresh skin tissue yielded positive results). In light of this, we quantified collagen glycation through fluorescence measurements (λex 370 nm/λem 440 nm). Further details regarding this methodology can be found in the "Materials and Methods" section of our study.

Comment 4

In the paper, published in International Journal of Nanomedicine 2017:12 3303–3314, “Combining nano-physical and computational investigations to understand the nature of “aging” in dermal collagen”, an age-related decrease in the Young’s modulus of the transverse fibril (from 8.11 to 4.19 GPa in young to old volunteers, respectively, P,0.001) was reported. Authors should discuss the discrepancy.

Response: We referenced and discussed above paper (see “Discission” for details).

Comment 5

In Figs 3c and 4c, the spelling of punch in “Young skin puch biopsy” should be corrected.

Response: The typos are corrected.

---

## [Editor Report · Decision Letter 1]

28 Sep 2023

Age-related changes in dermal collagen physical properties in human skin

PONE-D-23-19389R1

Dear Dr. Quan,

We’re pleased to inform you that your manuscript has been judged scientifically suitable for publication and will be formally accepted for publication once it meets all outstanding technical requirements.

Kind regards,

Nileshkumar Dubey

Academic Editor

PLOS ONE